# Role of lean leadership in the lean maturity—second-order problem-solving relationship: a mixed methods study

Arie Bijl,[1] Kees Ahaus,[1,2] Gwenny Ruël,[1] Paul Gemmel,[3] Bert Meijboom[4]

[1]Department of Operations, Faculty of Economics and Business, Erasmus University Rotterdam, Rotterdam, The Netherlands
[2]Department of Health Services Management & Organisation, Erasmus School of Health Policy & Management, Erasmus University Rotterdam, Rotterdam, The Netherlands
[3]Department of Innovation, Entrepreneurship and Service Management, Faculty of Economics and Business Administration, Ghent University, Gent, Belgium
[4]Department of Tranzo, Tilburg University, Tilburg, The Netherlands

**Correspondence to**
Prof. Kees Ahaus;
ahaus@eshpm.eur.nl

## ABSTRACT

**Objectives** To investigate the relationship between lean adoption and problem-solving behaviour in nursing teams, and to explore the practices of lean leaders on nursing wards to reveal how they can stimulate second-order problem-solving within their teams.

**Design** A mixed-methods retrospective multiple case study using semistructured interviews. Interview data were used to assess the level of lean maturity (based on a customised validated instrument) and the level of second-order problem-solving (based on scenarios). Within-case and cross-case analyses were employed to identify lean leadership practices.

**Setting** 14 nursing teams, with different levels of lean maturity, in a Dutch hospital.

**Participants** Three members of each nursing team were interviewed: the team leader, one nurse from the ward's core team for the lean-based quality improvement programme and one nurse outside the core team.

**Interventions** The nursing teams were in various phases of a lean-based quality improvement programme: 'The Productive Ward – Releasing Time to Care'.

**Results** A strongly significant positive relationship between lean maturity and second-order problem-solving was found: β=0.68, R²=0.46, p<0.001. Further, the results indicated a potential strengthening effect of lean leadership on this relationship. Seven lean leadership practices emerged from the data collected in a nursing ward setting: (1) convincing and setting an example; (2) unlocking individual and team potential; (3) solving problems systematically; (4) enthusing, actively participating and visualising; (5) developing self-managing teams; (6) sensing, as orchestrator, what is needed for change; and (7) listening, sharing information and appreciating. These practices have a strong link with transformational leadership.

**Conclusions** As lean matures, nursing teams reach a higher level of second-order problem-solving. In later stages, lean leaders increasingly relinquish responsibility by developing self-managing teams.

## INTRODUCTION

The earliest applications of lean thinking in the healthcare sector were more than a decade ago. Since then, lean has gained in popularity in the sector.[1] The primary goals of

### Strengths and limitations of this study

► The study fills a gap in knowledge as to whether more mature lean teams show a higher level of second-order problem-solving behaviour.

► In addition, it provides valuable insight into typical lean leadership practices and how these are linked to mainstream leadership theories.

► Future research attempting to measure lean maturity in healthcare environments may benefit from using the newly developed and validated scale that incorporates six items from Malmbrandt and Ahlström's instrument.

► Another strength of the study is its use of Gioia methodology to inductively analyse the lean leadership data.

► A limitation is that the nursing teams all came from one Dutch hospital, however, this choice had the benefit that it provided nursing teams with a range of lean maturities working under similar circumstances.

lean in healthcare have been to increase the quality of care and to increase efficiency.[2] To achieve these goals, most healthcare organisations have emphasised the application of lean tools to reduce direct waste, but neglected developing the problem-solving abilities of frontline employees.[3] This approach may have created some process improvements, but long-term hospital-wide benefits have rarely been achieved.[4]

To realise more of lean's potential, it is often suggested that structured problem-solving should be developed throughout the organisation to sustainably improve processes.[3] This is inspired by Liker's 4P model that identifies four aspects of lean: philosophy, process, people and problem-solving.[3] One well-known and effective approach to this is *second-order problem-solving*. This involves the in-depth questioning of work practices to uncover and remove the root causes of problems.[5] This approach is in sharp contrast with *first-order problem-solving* where problems

are resolved in an ad-hoc manner, while underlying causes remain.[6] A recent study by Gemmel *et al*[3] suggests that second-order problem-solving is more prevalent than first-order problem-solving in nursing teams with high levels of lean maturity. However, evidence from a larger sample of wards is needed to confirm this.

Earlier studies have suggested that team leaders could stimulate second-order problem-solving behaviour by nurses.[5] Team leaders are crucial in creating and sustaining the benefits of lean adoption on hospital wards since they can help to create a culture of continuous improvement, empower frontline employees and foster participation.[7–9] Although the literature often emphasises the importance of lean leadership, there are few empirical studies of lean leadership in healthcare.[9 10] Moreover, previous studies have not connected lean leadership with other leadership theories.[9] Consequently, an in-depth empirical study of lean leadership on nursing wards is needed, whereby the identified leadership practices can be positioned within existing leadership approaches in order to enhance our understanding of this concept.

In this paper, we study the relationship between lean maturity and second-order problem-solving behaviour. In addition, we explore the meaning attached to lean leadership on nursing wards and discover how lean leadership moderates the lean maturity—second-order problem-solving relationship. In our study of the lean leadership concept, we focus solely on identifying leadership practices on nursing wards (ie, in terms of team leadership). We answer the following research questions:

► How does lean maturity affect the problem-solving behaviour of nursing teams?
► What constitutes lean leadership on nursing wards?
► How does lean leadership affect the relationship between lean maturity and second-order problem-solving?

## BACKGROUND
### Lean maturity
The concept of lean maturity needs careful consideration if one is to accurately assess the impact of lean implementation on the problem-solving behaviour of nursing teams.[3] Malmbrandt and Åhlström[11] argue that the extent of an organisation's lean adoption can be measured using an instrument that incorporates measures to assess lean enablers, lean practices and performance. Lean enablers represent the supporting structure or preconditions of lean, including the training of employees and dedicating time and resources to improvement work. Lean practices correspond to lean principles, such as continuous improvement and eliminating waste. Performance refers to the results of lean adoption in terms of measures such as quality, customer satisfaction and costs. Together, these three dimensions determine the lean maturity level of an organisation. These can range from no adoption through the increase of continuous improvement activities to an exceptional, well-defined and innovative approach.[11]

### Second-order problem-solving behaviour
Nurses, as frontline service providers, play an essential role and are in the best position to uncover and eliminate the root causes of problems and, thereby, help their team learn.[5 6] Second-order problem-solving occurs when 'the worker, in addition to patching the problem-so that the immediate task at hand can be completed (ie, first-order problem-solving), also takes action to address underlying causes'. (Mazur and Chen, p63)[6]

Tucker and Edmondson (p61)[5] distinguish five broad actions linked to second-order problem-solving: (1) communicating to the person or department responsible for the problem; (2) bringing the problem to the manager's attention; (3) sharing ideas about the cause of the situation and how to prevent recurrence; (4) implementing changes; and (5) verifying that changes have the desired effect.

Nurses apply both first- and second-order problem-solving approaches. Quick workarounds may be needed if patients cannot wait for the promised care.[12] However, only by applying second-order problem-solving can real process improvements be made. Gemmel *et al*[3] go as far as to state that a learning organisation based on second-order problem-solving should be one of the ultimate goals of lean.

### Lean leadership
The importance of effective 'Lean leadership' during lean implementations is widely recognised.[8] A few studies in healthcare have suggested that lean leadership can be linked to transformational leadership theory.[9 13] Under transformational leadership, followers are motivated to do more than originally expected and to feel trust, loyalty, respect and admiration towards their leader.[14] Transformational leaders develop their followers through four key dimensions. *Idealised influence* refers to a leader being a role model, one who is admired and respected by their followers. *Inspirational motivation* implies that a leader motivates those around him/her and arouses their spirit. Transformational leaders use *intellectual stimulation* to stimulate their followers to be innovative and creative. Finally, a transformational leader attends to the need for achievement and growth of individuals through *individual consideration* by acting as a coach.[14] Transformational leadership is different from transactional leadership. The latter is merely an exchange process to motivate follower compliance, where a leader clarifies performance criteria, states expectations and determines what followers receive in return.[14] The link between transformational leadership theory and lean leadership is relatively under-researched.[9] In healthcare, there are only a few empirical studies investigating lean leadership practices.[9 10]

### Conceptual model
Figure 1 displays the conceptual model applied in this study. First, in line with Gemmel *et al*,[3] we expect greater lean maturity to potentially lead to an elevated degree of second-order problem-solving in nursing teams. Second, since lean leaders are seen as essential in enhancing the

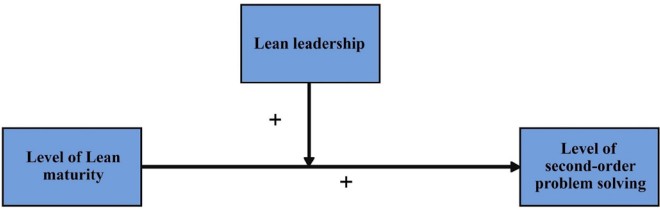

**Figure 1** Conceptual model.

problem-solving abilities of healthcare staff,[8] we expect effective lean leadership to strengthen the relationship between lean maturity and second-order problem-solving.

## METHODS

This research involves 14 case studies of nursing teams in different departments of a Dutch hospital, all in different phases of a lean-based quality improvement programme: 'The Productive Ward – Releasing Time to Care' (from now on referred to as PW). This programme trains nursing teams in how to apply lean tools and principles in their daily work and claims to increase the time nurses have for direct patient care.[15] A key tool incorporated in PW is the Hairdryer Model. This teaches nurses how to engage in effective problem-solving by discussing problems in groups, mapping the current situation, collecting data, implementing changes and assessing their impacts.

Each of the 14 nursing teams has a PW core team that leads the change. Generally, this consists of the team leader and a few nurses. To obtain reliable data, the team leader, one nurse inside and one nurse outside each PW core team were interviewed. The sample consisted of 7 males and 35 females; 14 of whom were employed for less than 5 years, 15 between 5 and 10 years, five between 10 and 15 years, one between 15 and 20 years and seven who had worked 20 or more years in the hospital. The nursing teams came from various wards: cardiology, urology orthopaedics, coronary care unit (CCU), emergency department, acute admission, neurology, lung medicine, birth centre, neonatology, paediatrics, oncology, short-stay and day nursing. Nursing teams with differing periods in the programme (0–24 months) were selected as cases. We expected this to provide us with variation on the lean maturity variable. Cases were selected with different durations in the belief this would produce different results but for predictable reasons (enabling theoretical replication).[16] Table 1 shows the selected cases within each duration period.

**Table 1** Selected cases within each duration period

| Duration | Selected cases |
| --- | --- |
| Period 1 (0–6 months) | L, N |
| Period 2 (6–12 months) | H, I, J, K |
| Period 3 (12–18 months) | D, E, F, G, M |
| Period 4 (18–24 months) | B, C |
| Period 5 (24+ months) | A |

### Data collection

In total, 42 semistructured interviews with nurses were conducted in December 2016 with an average duration of 40 min (minimum 24 min, maximum 68 min). More than one researcher was present at each interview. The interview questions related to lean maturity were designed to measure several lean enablers and lean practices. These provided accurate insight into the progress being made with lean implementation. These lean enablers and practices were drawn from Malmbrandt and Åhlström's instrument.[11] Their instrument contains 34 items to assess lean service adoption but we decided to include only six items for several reasons. First, some items had strong links with our other main variables, such as management commitment and understanding (lean leadership), the degree of structured problem-solving (second-order problem-solving) and performance and these were excluded to avoid confounding issues between the different concepts/variables of our conceptual model. Second, some items were not considered relevant in the hospital context, or too complex to measure (such as the levelling and balancing of workloads). These exclusions resulted in only six items being considered useful for this study. The customised instrument included three lean enablers (employees' understanding of lean; time and resources allocated to improvement work; and bi-directional vertical information flows), and three lean practices (identification of patient value, workplace design for flow; and visualisation of information and improvements).

Second-order problem-solving was measured through a set of questions in the form of scenarios derived by Gemmel *et al*.[3] These scenarios describe several types of problems nurses face in their daily work. They were used to obtain information about the actions nurses took when faced with a problem. To obtain further insights into this concept, nurses were also asked to provide examples of a problem where they engaged in second-order problem-solving.

Since the main goal of PW is to increase the proportion of time that nurses spend on direct patient care,[15] the level of performance is measured by an open question asking about perceptions of PW outcomes in terms of the time that is freed up for direct patient care.

Lean leadership was studied from various viewpoints, with different sets of open and semistructured interview questions posed to team leaders and nurses. We specifically asked about leadership practices adopted to support and implement the PW programme and to develop the team in applying PW. Questions related to generic leadership practices were avoided.

### Data analysis

From each interview, scores were deduced for the levels of lean maturity and second-order problem-solving behaviour based on the instruments applied. The scores for each set of three ward members were checked by the interviewers for consistency. This process eventually resulted, for each team, in a low, medium or high rating,

for its lean maturity and its second-order problem-solving level.

As an indicator of the construct validity of our instrument, a correlation analysis was performed to explore whether the duration of the PW programme was related to the level of lean maturity. This revealed a strong correlation (r=0.58; p<0.001). Next, scores were attributed to the three enablers and three practices of lean on a five-point scale based on the original instrument by Malmbrandt and Åhlström.[11] Because the lean enablers and practices were extracted from a validated instrument, a factor analysis was performed to check for unidimensionality of the six selected items. Since our sample size was too small (n=42) for a confirmatory factor analysis (the model fit indices: root mean square of the residuals (RMSR)=0.11, root mean square error of approximation (RMSEA)=0.16 and Tucker Lewis Index (TLI)=0.82 did not fully meet the criteria), a reasonable alternative consists of a two-step approach in which we first used principal component analysis (PCA) to provide evidence on the existence of only one dimension, and subsequently confirmed its internal consistency by means of a reliability analysis using Cronbach's alpha. With Kaiser Meyer Olkin measure (KMO)=0.76, Bartlett's chi-square=92.2; p<0.001, all communalities >0.49 and explaining 56.2% of the variance, conducting a PCA was allowed. The PCA extracted a single component (Eigenvalue factor 1=3.37; Eigenvalue factor 2=0.77) with all factor loadings above 0.7, supporting the construct validity of measuring lean maturity using the six-item scale. In addition, the reliability analysis returned a Cronbach's alpha of 0.84 suggesting a strong internal consistency among the lean enablers and practices.

The information collected through the scenarios was used to link each respondent to a category of first- or second-order problem-solving behaviour. For this, we used a scale with eight levels. Levels 1, 2 and 3 represent first-order problem-solving approaches, whereas levels 4 through eight represent second-order problem-solving actions as described by Tucker and Edmondson, (p61)[5] with level eight indicating the highest degree of second-order problem-solving.

Since second-order problem-solving is expected to enhance performance, a correlation analysis was performed to assess this relationship. The perceived performance was categorised using a four-level: (1) perception that there are no clear effects of PW; (2) perception that time is freed up for direct patient care, but only indirectly through a better organised ward; (3) perception that actual time is freed up for direct patient care and this is exemplified; and (4) time is demonstrably freed up for direct patient care.

For the within-case analysis of lean leadership, data were extracted from all the interviews. Using the rigorous step-by-step approach proposed by Gioia *et al*,[17] the data were coded inductively by two researchers. This led to a coding tree of lean leadership practices (available on request) involving 339 in vivo codes, 29 themes and seven aggregate dimensions. The list of themes was then used

by two researchers to independently code 9 out of 42 interviews (20% of the interviews). A comparison demonstrated an acceptably high inter-rater reliability in terms of assessing lean leadership with a Krippendorff's alpha value of 0.85.[18]

In the cross-case analysis, cases where consistent low, medium or high levels of both lean maturity and second-order problem-solving were found were compared with cases that deviated from this expectation of consistency to determine whether lean leadership strengthened the relationship between lean maturity and second-order problem-solving.

### Patient and public involvement

There was no patient or public involvement in this study. The results of this work were disseminated to representatives of the hospital. Once the study has been published, the main findings will be used in the education of Master students.

## RESULTS

The results showed that increased lean maturity positively influences the level of second-order problem-solving in nursing teams. This relationship was tested through a single linear regression analysis. A strongly significant positive relationship was found between lean maturity and second-order problem-solving: β=0.68, $R^2$=0.46, p<0.001. There was also a strong correlation between second-order problem-solving and performance (r=0.66; p<0.001), indicating that second-order problem-solving indeed seems to free up time for patient care.

### Within-case analysis

The in-depth analysis of the meaning of lean leadership resulted in the identification of seven lean leadership practices: (1) convincing and setting an example; (2) unlocking individual and team potential; (3) solving problems systematically; (4) enthusing, actively participating and visualising; (5) developing self-managing teams; (6) sensing, as orchestrator, what is needed for change; and (7) listening, sharing information and appreciating. Each of these practices is explained below.

First, lean leaders helped nurses to become acquainted with the programme. They constitute the driving force of PW by setting an example and actively promoting PW in their team. Initially, many nurses needed to be convinced of the value of PW and, so, an important activity was to explain PW modules and highlight potential benefits. These included the time that could be saved by applying PW: *'I try to give insights into why we have to change certain things'* (D-01: in this notation, D refers to team D, participant '01' refers to the team leader; '02' and '03' refer to the nurses inside and outside the PW core team respectively).

Lean leaders also aimed to unlock the greater potential of nurses and their team by encouraging everyone to become active within PW, for instance, by stimulating

membership of the PW core team and participation in improvement board sessions. Nurses were also invited to indicate a preferred improvement project, based on their own interests and capabilities, to work on within PW. Within these PW projects, lean leaders adjusted their own degree of involvement for each individual based on the level of experience of each nurse. This gave room for individual potential to develop.

Further, lean leaders were involved in systematic problem-solving. They were present during team discussions where problems were analysed by applying PW tools. Leaders also encouraged their team to solve problems through the Hairdryer Model, as they felt that this model could be incorporated in nurses' work practices. In addition, the leaders cooperated with nurses who wanted to engage in systematic problem-solving, as one of the participants stated: *'Our team leader would help us by providing input on how we can approach things'* (C-03).

Another key lean leadership practice was to enthuse the team, actively participate in PW and visualise. First, the leaders encouraged nurses to organise improvement board sessions and to carry out daily evaluations of processes. We observed that core-team members were regularly rotated, as being a member of the PW core team was perceived as a strong motivating factor by nurses. This leadership involvement was seen as highly important by both nurses and leaders, as the following quotes illustrate: *'The leader motivates me through her enthusiasm, and by showing that she wants it herself'* (N-02) and: *'I need to bring along the group through enthusiasm'* (E-01). Finally, many participants pointed out that leaders used visualisation through photographs or videos to create awareness of a non-optimal situation and to demonstrate PW accomplishments.

Lean leaders also developed self-managing teams. They encouraged nurses to actively take the lead, for instance, by giving nurses ownership of improving certain processes through PW tools. It was frequently noted that, as the implementation of PW progressed, lean leaders would increasingly relinquish tasks and responsibility to increase the sense of ownership among the nurses in the team. This interpretation is supported by the following quotes: *'Through this programme, my tasks are diminished as nurses themselves take more responsibility'* (A-01) and *'The leader does not want to keep tight control of everything, he relinquishes tasks and gives guidance, which motivates us to be actively involved in the programme'* (H-03).

Lean leaders continuously thought about what was needed for change to take place. For instance, they facilitated this by providing time and resources to engage in PW activities and they ensured that PW meetings regularly took place. It was also stated that the leaders provided direction and kept an overview of PW's progress within their team, without losing track of the hospital's objectives. The leaders focused on small and manageable steps for improvement within the PW modules, carefully assessing any resistance to PW from the team and giving constant attention to ensuring the changes took place.

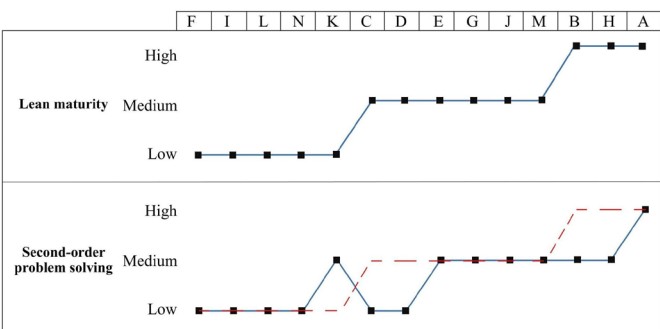

**Figure 2** Lean maturity levels and expected (dashed in red) and actual (blue) second-order problem-solving level per case/team.

Finally, lean leaders took time to listen, share information and appreciate the efforts of others. They engaged in individual conversations with nurses in order to stay involved and, if there were issues within the team, to better understand why these had emerged. Information about PW's progress was also shared actively with the team using score charts, newsletters and emails: *'It is my responsibility to inform them properly'* (K-01). Lean leaders also expressed appreciation to their team. Compliments were given if the team was on schedule and when certain modules had been completed successfully.

### Cross-case analysis

Figure 2 displays the levels of lean maturity and second-order problem-solving of each team. The blue lines represent the assessed levels for each team based on the interviews, while the red line shows the expected level of second-order problem-solving, based on the proposition that higher lean maturity results in increased second-order problem-solving.

The graph shows that this proposition did not hold for all the teams. In searching for patterns and reasons, cases that agreed with the proposition were compared with cases that deviated. This analysed whether the difference between actual and expected second-order problem-solving levels could be attributed to the practices of lean leaders. More specifically, teams C and D (medium lean maturity–low second-order problem-solving that is, 'ML') were compared with cases E, G, J, M (medium lean maturity–medium second-order problem-solving, ie, 'MM'). Further, cases B and H (high lean maturity–medium second-order problem-solving, ie, 'HM') were compared with team A (high lean maturity–high second-order problem-solving, ie, 'HH').

Our analysis shows that lean leaders are able to contribute to their team achieving a high level of second-order problem-solving. Second-order problem-solving can be increased by leadership practices that encourage team ownership of the problem-solving process. In the HH case, team A, the leader played a crucial role in empowering and stimulating nurses to take the lead in thinking about solutions to problems and in implementing solutions through a bottom–up approach. This was explained as follows: *'She [the team leader] tells me to go and investigate*

*how I could make a solution happen'* (A-03). In comparison, in the HM cases, the teams were self-managed to some extent, but responsibility for the entire problem-solving process had not been completely relinquished by the leaders. That is, in the latter, more involvement and active support of the leaders in the problem-solving process was observed.

Second, our analysis shows that an enthusiastic leader who actively participates in the lean initiative as a role model can lead to a higher level of second-order problem-solving. In the HH and MM teams, the leaders were seen as role models for PW through their enthusiasm and participation, as exemplified by the following quote: *'Our leader is very enthusiastic and fanatical'* (E-02). This was perceived as an important aspect that keeps PW alive in the team and, thereby, keeps nurses motivated to participate. In contrast, in the HM and ML cases, the level of enthusiasm and participation from leaders was perceived as being lower.

## DISCUSSION

The first main research question posed in this research was: *How does lean maturity influence the problem-solving behaviour of nursing teams?* Our findings confirm the related proposition that, as nursing teams reach higher levels of lean maturity, they also demonstrate higher degrees of second-order problem-solving. As such, we add to the evidence for the existence of a positive relationship between lean maturity and second-order problem-solving.[3] This suggests that nurses may become more skilled in discovering and removing the root causes of organisational problems through lean adoption.[19 20] The reason is that lean helps nurses to become acquainted with and involved in the identification, analysis and removal of the root causes of problems. In the PW programme, nursing teams were trained to use the Hairdryer Model that involves multiple activities associated with second-order problem-solving actions.[5]

The second research question was: *What constitutes lean leadership on nursing wards?* In order to answer this question, lean leadership was studied empirically in several nursing teams, resulting in the identification of seven lean leadership practices as listed in the within-case analysis section. Overall, our findings indicate that lean leadership on nursing wards has strong connections with transformational leadership theory, a finding in line with earlier studies.[9 13] That is, many of the identified lean leadership practices can be classified as transformational. For example, arousing the team spirit through motivation and inspiration are key activities in the inspirational motivation dimension.[14] Charismatic-inspirational leadership was also in evidence when lean leaders enthused others and, at the same time, actively participated in the lean programme themselves.[21] A further connection with individualised consideration and coaching to develop followers was observed in the lean leaders' practice of unlocking individual and team potential. This was achieved by recognising differences in individuals' qualities

and preferences within the team.[21] Our results indicate that lean leaders also used other transformational practices, such as listening effectively, actively sharing information and appreciating people.[21] Our results also underpin the importance of visualisation in applying lean in hospital environments.[22] Visualisation was practised by lean leaders to help their team identify areas for improvement[9] and to demonstrate PW accomplishments on improvement boards that were located on the nursing wards.

The final research question was: *How does lean leadership affect the relationship between lean maturity and second-order problem-solving?* The cross-case analysis revealed that successful lean leaders increasingly relinquished responsibility for improvement activities to their team as lean maturity increased. At the outset of lean initiatives, large investments from leaders were required in terms of effort and resources[9] and top-down steering may initially be necessary to create the supporting structure for lean.[23] However, in later stages, it becomes imperative to empower frontline staff so that they can take the initiative in daily improvement activities.[10] Our findings suggest that such a transition was necessary for nursing teams to reach a high level of second-order problem-solving. Lean leaders have an important role in facilitating this transition by encouraging nurses to take ownership of improvement activities and by developing their team to become self-managing. The importance of stimulating a bottom–up approach to continuous improvement in healthcare is supported by existing literature.[20 22]

Overall, our findings suggest that successful lean implementation in nursing teams requires a bottom–up approach with responsibility for improvement activities gradually being handed over to frontline nurses, supported by leaders with strong transformational leadership skills that radiate enthusiasm towards their team. Through this, nursing teams can achieve a high second-order problem-solving level as lean matures.

This study has a number of limitations. First, since nursing teams from only one Dutch hospital were included, the external validity of our findings could be impacted negatively. However, this choice had the benefit that it enabled us to compare nursing teams with a range of lean maturities working under similar circumstances. Second, when we started the interview round, we realised that many respondents did not recognise all the problems provided in the scenarios of Gemmel *et al*.[3] This was counteracted at an early stage of data collection by asking an additional open-ended question as to whether nurses could provide an example of a problem-solved through a second-order problem-solving approach. This resulted in many useful responses.

Further research could aim to improve the instrument for measuring second-order problem-solving in nursing teams. This could involve adding new scenarios in which nurses are able to recognise their problem-solving behaviour. Other research that wants to measure lean maturity could benefit from adopting the validated scale, comprising six items from Malmbrandt and Ahlström's

more detailed instrument,[11] that this study found to be relevant and measurable on nursing wards.

**Acknowledgements** We would like to thank Iris Brouwer, Pauline Vinks and Thomas Fransen for their contributions to the design, data collection and data analysis of the study. We are further grateful to the study participants from the hospital.

**Contributors** AB contributed to the conception and design of the study, data analysis, interpretation of the data and in drafting the manuscript. AB and KA analysed the lean leadership interview data. KA and GR contributed to the study's conception and design, and to revising the paper. PG and BM shared the scenarios from a prior study and contributed to revising the paper. KA coordinated the study. Finally, all authors read and approved the submitted manuscript.

**Funding** This research received no specific grant from any funding agency in the public, commercial or not-for-profit sectors.

**Competing interests** None declared.

**Provenance and peer review** Not commissioned; externally peer reviewed.

**Data sharing statement** No additional data are available.

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
