## [Reviewer comments · BMJ Open]

ARTICLE DETAILS

TITLE (PROVISIONAL)	The role of lean leadership in the lean maturity – second-order problem-solving relationship: a mixed methods study
AUTHORS	Bijl, Arie; Ahaus, Kees; Ruël, Gwenny; Gemmel, Paul; Meijboom, Bert

VERSION 1 - REVIEW

REVIEWER	Theresa Kline University of Calgary Canada
REVIEW RETURNED	23-Oct-2018

GENERAL COMMENTS	This is a clearly-written manuscript that describes the results of a qualitative research study regarding problem-solving development and the role of leadership in that development. The link to extant leadership literature and acknowledgment of some of the limitations of the study were clearly outlined. It would be helpful to describe more clearly why only 6 of the 34 of the questions for the Lean service adoption were asked. This is particularly important from the performance end. That is, what impact did the second order problem-solving have on outcomes? This, presumably, is the end goal of the entire process, yet it was not included. There needs to be some rationale for doing so.
---

REVIEWER	Rubin Cohen Amarillo VA Health System Amarillo, TX USA
REVIEW RETURNED	24-Nov-2018

GENERAL COMMENTS	A qualitative study on Lean implementation. The authors studied mature Lean teams and evaluated second-order problem solving as well as the impact of coaching, facilitating and leadership. The study was performed in one Dutch hospital, thus the results may not apply to other hospitals or other healthcare settings. Nevertheless, the study illustrates that Lean is a continuous learning process
--

REVIEWER	Maria Engström University of Gävle, Department of Health and Caring Science, Sweden
REVIEW RETURNED	26-Dec-2018

GENERAL COMMENTS	Thank you for the opportunity to review this manuscript. The manuscript covers an important and interesting topic and the results add to current research finding. Introduction and Background: The introduction is, in general, well written. However, I miss a section describing lean, different definitions/views of Lean and the definition of Lean used in the present study. Some definitions of Lean include problem solving. Methods: Please, describe the sample in more detail (e.g. gender, age, years worked in the current position...) what kind of wards surgical, medical wards? Based on the comment above (introduction/background) is Lean maturity measured or some Lean enablers and practices? The interviews where according to the text (page 5) coded by multiple researchers to enhance the reliability, please describe results from this. Further, "The scores for each set of three ward members were checked for consistency", please present results. According to text page 5 a CFA was preformed (please consider the accuracy of using a CFA with only 43 participants and present the theoretical assumption), present values for fit indices. Results: The results are interesting and add to current research. Title: please consider to revise the title to better capture the mixed methods approach used. As presented the reader expect to find quantitative results of lean leaders as a moderator in the explored relationship between lean maturity and second order problem-solving. However, this part is based on qualitative data and is more explorative and describe how lean leadership is practiced in different cases. Abstract: According to the abstract the interview data were supplemented by document research, what was that? Please, be clearer regarding results from the mixed methods approach.
---

VERSION 1 – AUTHOR RESPONSE

Reviewer 1	
3	This is a clearly-written manuscript that describes the results of a qualitative research study regarding problem-solving development and the role of leadership in that development.
Response	Thank you.
4	It would be helpful to describe more clearly why only 6 of the 34 of the questions for the Lean service adoption were asked
Response	We apologize for not explaining clearly why we used only 6 of the 34 questions. Following your suggestions we have clarified the text as follows:

	'These Lean enablers and practices were drawn from Malmbrandt and Åhlström's instrument.[1] Their instrument contains 34 items to assess Lean service adoption but we decided to include only six items for several reasons. First, some items had strong links with our other main variables, such as management commitment and understanding (Lean leadership), the degree of structured problem-solving (second-order problem-solving) and performance, and these were excluded to avoid confounding issues between the different concepts/variables of our conceptual model. Secondly, some items were not considered relevant in the hospital context, or too complex to measure (such as the levelling and balancing of workloads). These exclusions resulted in only six items being considered useful for this study. The customized instrument included three Lean enablers (employees' understanding of Lean; time and resources allocated to improvement work; and bi-directional vertical information flows), and three Lean practices (identification of patient value; workplace design for flow; and visualization of information and improvements).'
5	This is particularly important from the performance end. That is, what impact did the second order problem-solving have on outcomes? This, presumably, is the end goal of the entire process, yet it was not included.
Response	We fully agree that the end goal of PW is performance improvement in terms of freeing up more time for patients. In the study, we intended to make use of multi-moment analysis data on the time spent by nurses on direct patient care. However, the data were lacking or incomplete in most cases, forcing us to exclude performance. However, we do have data on perceived performance and therefore, based on your suggestion, we have added the following text in the data collection paragraph: 'Since the main goal of PW is to increase the proportion of time that nurses spend on direct patient care[16], the level of performance is measured by an open question asking about perceptions of PW outcomes in terms of the time that is freed up for direct patient care.' Further, we added the following to the data analysis paragraph: 'Since second-order problem solving is expected to enhance performance, a correlation analysis was performed to assess this relationship. The perceived performance was categorized using a four-level scale: (1) perception that there are no clear effects of PW; (2) perception that time is freed up for direct patient care, but only indirectly through a better organized ward; (3) perception that actual time is freed up for direct patient care and this is exemplified; and (4) time is demonstrably freed up for direct patient care.' Finally, we added in the Results section: 'There was also a strong correlation between second-order problem-solving and performance ($r=0.66$; $p<.001$), indicating that second-order problem-solving indeed seems to free up time for patient care.'
Reviewer 2	
6	The study was performed in one Dutch hospital, thus the results may not apply to other hospitals or other healthcare settings.
Response	We fully agree with the reviewer. In the paper, we acknowledge that since nursing teams from only one Dutch hospital were included, the external validity of our findings are unproven. Although this could be seen as a limitation, a benefit was that it provided a sample of 14 nursing teams with a range of Lean maturities working under similar circumstances. The uniqueness of the current study is that it is an in-depth exploration of this relationship that includes new and untested variables (such as leadership).

7	Nevertheless, the study illustrates that Lean is a continuous learning process.
Response	Indeed, because nursing teams with a range of Lean maturities participated, we could demonstrate that second-order problem-solving evolves as lean maturity grows since all the nursing teams worked under broadly similar circumstances within the same hospital.
Reviewer 3	
8	The manuscript covers an important and interesting topic and the results add to current research finding.
Response	Thank you.
9	I miss a section describing lean, different definitions/views of Lean and the definition of Lean used in the present study. Some definitions of Lean include problem solving.
Response	We agree with the reviewer that Lean definitions can include problem solving. We incorporate this aspect and refer to the seminal work of Liker (2004). We have added the following text in the introduction: ‘This is inspired by Liker’s 4P model that identifies four aspects of Lean: philosophy, process, people, and problem-solving.[5]’
10	Please, describe the sample in more detail (e.g. gender, age, years worked in the current position...) what kind of wards surgical, medical wards?
Response	All respondents were nurses: the team leader, one nurse who was a member of the core team, and one nurse from outside each PW core team. Unfortunately, we do not have information on their ages. Given that our independent variable was lean duration, age seemed to be less important at the time of data collection. We have added extra text with respect to gender, working years and type of wards: ‘The sample consisted of 7 males and 35 females; 14 of whom were employed for less than five years, 15 between five and ten years, 5 between ten and fifteen years, 1 between fifteen and twenty years and 7 who had worked twenty or more years in the hospital. The nursing teams came from various wards: cardiology, urology, orthopaedics, CCU, emergency department, acute admission, neurology, lung medicine, birth centre, neonatology, paediatrics, oncology, short-stay and day nursing.’
11	Based on the comment above (introduction/background) is Lean maturity measured or some Lean enablers and practices?
Response	Lean maturity is indeed measured in terms of lean enablers and lean practices, based on Malmbrandt and Åhlström[1]. In their article, they argue that the extent of an organization’s Lean adoption can be measured using an instrument that incorporates measures to assess Lean enablers. Lean practices and performance. In our study, Lean performance was excluded since performance was treated as a separate concept/variable. As a check on the validity of this, a correlation analysis was performed to explore whether the duration of the PW programme corresponded with the level of Lean maturity. This revealed a strong correlation ($r = 0.58$; $p < 0.001$), suggesting our instrument had construct validity. We added the text: ‘As an indicator of the construct validity of our instrument, a correlation analysis was performed to explore whether the duration of the PW programme was related to the level of Lean maturity. This revealed a strong correlation ($r = 0.58$; $p < 0.001$).’

12	The interviews were according to the text (page 5) coded by multiple researchers to enhance the reliability, please describe results from this.
Response	Thank you for this question. The sentence 'The interviews were coded by multiple researchers to enhance the reliability of the findings.[17]' was redundant as this point was repeated by stating 'the scores for each set of three members were checked for consistency'. Thus, we removed this sentence, and the text now reads as follows: 'From each interview, scores were deduced for the levels of Lean maturity and second-order problem-solving behaviour based on the instruments applied. The scores for each set of three ward members were checked by the interviewers for consistency. This process eventually resulted, for each team, in a low, medium or high rating for its Lean maturity and its second-order problem-solving level.'
13	Further, "The scores for each set of three ward members were checked for consistency", please present results.
Response	Unfortunately, we are unable to present these results. The interviewers scored individually and discussed the marking where scores differed. This led to consensus on scores for the levels of Lean maturity and second-order problem solving behaviour. The results of that assessment are depicted in figure 2.
14	According to text page 5 a CFA was preformed (please consider the accuracy of using a CFA with only 43 participants and present the theoretical assumption), present values for fit indices.
Response	Thank you for your comment. We should have explained whether it was allowed to perform a CFA. According to Malhotra (1999; p. 588-600) data have to fit for a CFA in general. To test this, KMO should be $>.5$; and Bartlett's chi-square test should be significant; all communalities should be $>.4$; and explained variance should be $> 60\%$. Only the explained variance was just below 60%, suggesting a possible second dimension, but since the second dimension had an Eigenvalue of .76 (whereas the first dimension had an Eigenvalue of 3.3) we decided to adhere to one dimension only. Therefore we added in the text: 'Since we used an adapted instrument and our sample size was small (N=42) we first looked whether the data fitted a CFA. The KMO = .76 and Bartlett's chi-square = 92.2; $p < .001$ with all communalities $> .49$ and explaining 56.2% of the variance, which allowed for a CFA.' Malhotra, N.K.. Marketing Research: An applied Orientation. Prentice-Hall International
15	The results are interesting and add to current research.
Response	Thank you.
16	Please consider to revise the title to better capture the mixed methods approach used. As presented the reader expect to find quantitative results of lean leaders as a moderator in the explored relationship between lean maturity and second order problem-solving.
Response	The reviewer touches upon the use of the term moderator, which might indicate a more quantitative approach to a moderating role. As we consider lean leadership to be a nascent field, we opted for an explorative approach to the interview data. To avoid given misleading expectations, we have followed the reviewer's suggestion and revised the title to: The role of lean leadership in the lean maturity – second-order problem-solving relationship: a mixed methods study.

17	According to the abstract the interview data were supplemented by document research, what was that? Please, be clearer regarding results from the mixed methods approach.
Response	In the PW programme, teams are asked to reflect on the suggested lean interventions. In addition, time-based studies are undertaken and analysed (e.g. multi-moment analysis) as part of the programme to free-up time for patients by reducing waste. Documents with results and progress plus short questionnaires in the PW programme were available for triangulation. However, we did not undertake an in-depth document study, and have therefore removed this suggestion from the abstract.

VERSION 2 – REVIEW

REVIEWER	Maria Engström Faculty of Health and Occupational Studies University of Gävle, Sweden
REVIEW RETURNED	08-Feb-2019

GENERAL COMMENTS	The paper has improved and most of my comments have been addressed. The paper is well written and the results are interesting and add to current research. Regarding the confirmatory factor analysis I still think the sample is too small for a CFA. I still also miss values/results from the goodness-of-fit indices. What kind of factor analysis has been performed?
--

VERSION 2 – AUTHOR RESPONSE

No	Comments and answers to comments
Reviewer 3	
1	The paper has improved and most of my comments have been addressed. The paper is well written and the results are interesting and add to current research. Regarding the confirmatory factor analysis I still think the sample is too small for a CFA. I still also miss values/results from the goodness-of-fit indices. What kind of factor analysis has been performed?
Response	We thank the reviewer for this comment which helped us to improve the substantiation of the validity and reliability of the six-item scale for lean maturity. The reviewer refers to the following text: Although the Lean enablers and practices were extracted from a validated instrument, a confirmatory factor analysis (CFA) was performed to test the unidimensionality of the six selected items. Since we used an adapted instrument and our sample size was small (N=42) we first looked whether the data fitted a CFA. The KMO = .76 and Bartlett's chi-square = 92.2; $p < .001$ with all communalities $> .49$ and explaining 56.2% of the variance, which allowed for a CFA. The CFA extracted a single component with all factor loadings above 0.7, indicating the construct validity of measuring Lean maturity using our

No	Comments and answers to comments
	six-item scale. In addition, the reliability analysis returned a Cronbach's alpha of 0.84 suggesting a strong internal consistency among the Lean enablers and practices. The reviewer specifically asks for reflecting on sample size and for adding goodness-of-fit indices and type of factor analysis. The revised text now reads as: Because the Lean enablers and practices were extracted from a validated instrument, a factor analysis was performed to check for unidimensionality of the six selected items. Since our sample size was too small (N=42) for a confirmatory factor analysis (the model fit indices: RMSR = 0.11, RMSEA = 0.16, and TLI = 0.82 did not fully meet the criteria), a reasonable alternative consists of a two-step approach in which we first used principal component analysis (PCA) to provide evidence on the existence of only one dimension, and subsequently confirmed its internal consistency by means of a reliability analysis using Cronbach's alpha. With KMO = 0.76, Bartlett's chi-square = 92.2; $p < .001$, all communalities > 0.49, and explaining 56.2% of the variance, conducting a PCA was allowed. The PCA extracted a single component (Eigenvalue factor 1 = 3.37; Eigenvalue factor 2 = .77) with all factor loadings above 0.7, supporting the construct validity of measuring Lean maturity using the six-item scale. In addition, the reliability analysis returned a Cronbach's alpha of 0.84 suggesting a strong internal consistency among the Lean enablers and practices.

VERSION 3 - REVIEW

REVIEWER	Maria Engström Department of Caring Science Faculty of Health and Occupational Studies University of Gävle, Sweden
REVIEW RETURNED	15-Mar-2019

GENERAL COMMENTS	The issue I raised have been fully-addressed and I look forward to seeing the paper in print.
---